# Cell-Free DNA Fragmentomics: A Promising Biomarker for Diagnosis, Prognosis and Prediction of Response in Breast Cancer

**DOI:** 10.3390/ijms232214197

**Published:** 2022-11-17

**Authors:** Caterina Gianni, Michela Palleschi, Filippo Merloni, Giandomenico Di Menna, Marianna Sirico, Samanta Sarti, Alessandra Virga, Paola Ulivi, Lorenzo Cecconetto, Marita Mariotti, Ugo De Giorgi

**Affiliations:** 1Department of Medical Oncology, IRCCS Istituto Romagnolo per lo Studio dei Tumori (IRST) “Dino Amadori”, 47014 Meldola, Italy; 2Biosciences Laboratory, IRCCS Istituto Romagnolo per lo Studio dei Tumori (IRST) “Dino Amadori”, 47014 Meldola, Italy

**Keywords:** breast cancer, fragmentomics, biomarkers, liquid biopsy, cell-free DNA, cell-free DNA integrity

## Abstract

Identifying novel circulating biomarkers predictive of response and informative about the mechanisms of resistance, is the new challenge for breast cancer (BC) management. The integration of omics information will gradually revolutionize the clinical approach. Liquid biopsy is being incorporated into the diagnostic and decision-making process for the treatment of BC, in particular with the analysis of circulating tumor DNA, although with some relevant limitations, including costs. Circulating cell-free DNA (cfDNA) fragmentomics and its integrity index may become a cheaper, noninvasive biomarker that could provide significant additional information for monitoring response to systemic treatments in BC. The purpose of our review is to focus on the available research on cfDNA integrity and its features as a biomarker of diagnosis, prognosis and response to treatments in BC, highlighting new perspectives and critical issues for future applications.

## 1. Introduction

Breast cancer (BC) is the most prevalent cancer worldwide, with at least 7.8 million patients diagnosed with BC in the last five years [1]. Thanks to new therapies, survival and the possibility of cure have considerably increased in recent times [2]. Disease monitoring has become a priority for millions of patients, and a social issue concerning the proper use of scientific and financial resources. In particular, the identification of novel serum biomarkers, less invasive than standard methods based on tumor tissue analysis, is a real challenge for BC detection and prediction of response to treatments. In this respect, liquid biopsy is revolutionizing clinical practice as a non-invasive diagnostic tool [3]. Applications of liquid biopsy in BC are copious in various settings, from diagnosis (identifying actionable molecular alteration and predictive biomarkers) to treatment, with the objectives of predicting outcome, detecting early progression, causing minimal residual disease, acting as a mechanism of resistance and focusing on treatment management (i.e., tailoring treatment duration, escalation/de-escalation strategies) [3].

Biomarkers detectable using liquid biopsy are many and varied, such as circulating tumor cells, inflammatory cells, mRNA, extracellular vesicles (i.e., exosomes), micro-RNA, circulating DNA, and more [4,5]. The combined analysis of circulating tumor cells and circulating DNA may offer a new approach for the monitoring of disease progression and for individualizing therapy as escalating/de-escalating approaches and/or targeted therapies in patients with advanced BC [6,7,8,9]. In particular, circulating tumor DNA (ctDNA) has emerged as the most useful circulating tool in BC [10]. CtDNA can overcome the bias of analysis in tissue sampling that does not represent tumor heterogeneity, especially in advanced disease, and can detect minimal residual disease or facilitate early diagnosis [11,12]. However, while advanced BC patients may present higher levels of ctDNA (it is detectable in >90% patients with metastatic cancer), it is rarely detectable in early BC [12]. Recent improvements in more sensitive technologies allowed the detection of ctDNA also in early-stage disease, but the application in standard clinical practice is significantly lacking [13,14]. One of the major concerns about ctDNA is the use of mutation-based detection assays (i.e., single-nucleotide variant, somatic copy number alterations, gene amplifications or methylation) particularly useful for differentiating ctDNA from cell-free DNA (cfDNA) [15,16]. However, most ctDNA in blood does not present any mutations, particularly in the earliest stages of the disease, which explains the difficulties of the application in these precise conditions and the low capacity for anticipating the diagnosis of localized cancer [3]. From this point of view, different approaches directed towards epigenetic analysis may be reasonable tools to fully exploit liquid biopsy in various settings.

The examination of fragmentation patterns of cfDNA, also known as “fragmentomics”, has been shown to be a valuable method to extract information from liquid biopsy without the need for mutational information, and it is now an attractive area of cancer biomarker research [17].

## 2. CfDNA Fragmentomics in BC: Different Features and Applications 

CfDNA in the bloodstream (and also in other body fluids) consists of DNA fragments released from cells undergoing apoptosis (in particular hematopoietic cells via caspase-dependent DNA cleavage). The enzymatic cleavage of nucleosome units in normal cells produces DNA fragments of at least 200 base pairs (bp) [18]. Every fragment is constituted of a DNA strand rolled up around nucleosomes (147 bp) and smaller DNA fragments of approximately 20 bp that act as linkers [19]. In the case of cancer cells, many different death processes are involved, such as necrosis, apoptosis, autophagy, necroptosis or ferroptosis [20]. These various events release DNA fragments of different sizes as compared to normal cells; however, long and short fragments may be released by the same mechanisms (e.g., necrosis and apoptosis) [21]. Moreover, cfDNA fragment characteristics and size could be influenced by secretion mechanisms, enzymatic activity, or the preservation process, and also by the tissue of origin [15,22]. 

The study of fragmentomics includes not only fragment size, but also other properties of cfDNA such as fragment endpoints, specific motifs included in the sequence of the fragment, the nucleosome footprints or position, and the various topologic forms (circular vs. linear) (Figure 1) [17]. 

### 2.1. Size of cfDNA Fragments and cfDNA Integrity

The first method of determining the size of cfDNA molecules was gel electrophoresis, subsequently replaced by quantitative polymerase chain reaction (qPCR) assay, able to define different fragment-size ranges by using amplicons with various measures [23]. The ratio between shorter and longer fragments is defined as the cfDNA integrity index (cfDI), and gives information about the total amount of tumoral DNA fragments. It is generally hypothesized that DNA fragments in plasma produced by cancer cells are more variable in length than cfDNA from non-cancer cells; however, this is not really been made clear so far, and the debate is open [24]. 

Most studies in BC have selected the ALU (Arthrobacter luteus) sequences, as they are considered the most frequent and repeated sequences in the human genome, and are commonly approximately 300 nucleotides in length, coinciding with more than 10% of the genome. The DNA integrity index corresponds to the ratio of ALU247/ALU115 [23]. Because the recombining sites of ALU115 are within the ALU247 annealing sites, the ratio of ALU247 to ALU115 is defined by DNA integrity. CfDI characterizes the fragmentation pattern of circulating cfDNA. In these studies, larger fragments, (such as ALU247), are assumed to derive from necrosis/autophagy of cancer cells, whereas shorter fragments (such as ALU115) derive from normal cells [21,25,26]. However, considerations of the actual length of tumor-derived cfDNA fragments are contradictory, and some studies enrolling cancer patients showed the exact opposite [27,28].

In BC, the qPCR technique has been the most widely used in multiple observational studies (Table 1). This method demonstrated that cfDI may easily detect early BC, showing a major level of cfDNA and a higher level of cfDI in BC patients versus healthy controls [26,29].

In this view, the use of cfDNA and cfDI may become in the future a feasible tool to improve early detection of BC and accuracy in cancer diagnosis, according to the cancer stage, after more confirming studies [30]. The analysis conducted by Umetani et al. on serum with a qPCR technique showed that the cfDI corresponding to the ratio ALU 247/115 was significantly associated with the stage at diagnosis, with an increase from stage II to stage IV [26]. Further evidence, using the same method, has been demonstrated by identifying high cfDI as an independent prognostic factor at the time of early BC diagnosis [31]. Iqbal et al. confirmed the correlation between the cfDI value and tumor size, also predicting the overall survival (OS) at 5 years and disease-free survival (DFS) at 4 years using the multivariate analysis [32]. In the same study, the observation of the cfDI trend during BC treatment revealed a significant decrease in DNA integrity in early BC, after surgery [32].

The size distribution of cfDNA and the cfDI variations have been described also under adjuvant/neoadjuvant treatment [33,34,35]. Wang et al. observed that the cfDI increased after neoadjuvant treatment in twenty-nine locally advanced BC patients as a consequence of the tumor shrinkage; in particular, patients undergoing complete pathological response presented a higher cfDI than patients with distant disease-progression after surgery [33]. However, in a similar study that included 65 patients with BC treated with neoadjuvant therapy, the cfDI was not informative, although the data were collected according to the same formulas as Umetani and Wang [36]. In the study conducted by Lehner et al., the absolute levels of ALU 115 showed a downward trend in patients with pCR, whereas they were increased in patients with disease stability. The same trend was observed with longer fragments (ALU 247), confirming other previous findings described for different histologies [36]. This downward trend was observed also by Adusei et al. in 32 female BC patients after three cycles of neoadjuvant chemotherapy [34]. The authors assumed that the reduction of cfDNA and cfDI after three cycles of chemotherapy was due to the decrease in tumor volume and to an improvement in cfDNA clearance; however, there are no clear data demonstrating that these assumptions might be true. In a study involving 41 BC patients in the adjuvant setting, a different variation of the two items was observed in two different groups of patients, independently of the type of treatment received [35]. One group showed an increase in the total amount of cfDNA and a corresponding rise in the cfDI, while the other subgroup revealed the opposite trend.

In the metastatic setting, Cheng et al. and Madhavan et al. reported that high cfDI was correlated as being related to better OS and RFS, in total opposition to results achieved in the early setting [32,37,38]. They used different ALU sequences (ALU 260 and ALU 111bp), and introduced another DNA recurring-element target, LINE-1 (LINE-1 266 and LINE-1 197 bp) [37,38]. Furthermore, Madhavan et al. rejected a possible prognostic value of cfDI in BC patients with non-metastatic disease who were represented in the study by a smaller cohort of 82 cases, as compared to 201 samples from advanced BC patients [37]. Cheng et al. also, surprisingly, observed that cfDI was lower in BC patients developing recurrence (*p* < 0.001 for ALU and LINE1 cfDI), hypothesizing that a reduction in cfDI could help define the risk of recurrence in association with clinical–biological features [38]. The body of evidence regarding cfDNA quantity and the application of cfDI is still lacking, because of the demonstration of opposite results and the presence of too much hypothesizing, rather than effective data, but the possible application to breast cancer diagnosis and monitoring is intriguing.

**Table 1 ijms-23-14197-t001:** Studies conducted into fragment size and cfDI in BC with qPCR techniques.

Study	Objective	BC Patients Enrolled (n)	Results or Status
NCT03474016	Diagnosis	116	Unknown
Agostini et al., 2012 [29]	Diagnosis	39	cfDI (ALU 247/125 ratio) was an accurate circulating biomarker for BC diagnosis (*p* < 0.0001).
Stötzer et al., 2014 [39]	Diagnosis	154	cfDNA was a valuable biomarker for the detection of localized BC.
Iqbal et al., 2015 [22]	Diagnosis	148	The ALU 247/115 ratio was significantly higher in BC patients, compared with controls (*p* < 0.001) and related to stage; cfDI trend during BC treatment revealed a significant decrease in eBC after surgery.
Kamel et al., 2016 [30]	Diagnosis	95	Higher cfDI values were useful for BC diagnosis (*p* < 0.001).
Elhelaly et al., 2022 [40]	Diagnosis	50	cfDNA and cfDI were markers for early diagnosis of BC, cfDNA concentrations were significantly lower after BC surgery (*p* < 0.001).
Hussein et al., 2019 [25]	Diagnosis	40	High cfDNA and cfDI (ALU 247/125 ratio) are diagnostic and preoperative prognostic markers for BC.
Umetani et al., 2016 [26]	Prediction of risk of relapse	83	cfDI may easily detect early BC and cfDI trend increase with cancer stage.
Maltoni et al., 2017 [41]	Prediction of risk of relapse	79	No association between cfDNA and prognosis, amplicones ratio between some longer and shorter fragments is higher in BC.
Cheng et al., 2017 [42]	Prediction of risk of relapse	212 (175 nonrecurrent, 37 recurrent BC)	ALU260/111 and LINE1-266/97 were lower in recurrent BC vs. nonrecurrent BC (*p* < 0.001). Lower cfDI BC patients may develop much more recurrence, compared with patients with higher cfDI (*p* = 0.020, *p* = 0.019).
Lamminhao et al., 2021 [31]	Prognostic value	207	High cfDI was an independent prognostic factor for poor OS in BC patients (*p* = 0.020).
Adusei et al., 2021 [34]	Variation of cfDI during neoadjuvant therapy	32	The study showed a downward trend of cfDNA and cfDI after three cycles of neoadjuvant chemotherapy in BC patients.
Wang et al., 2019 [33]	Description of variation of cfDI during neoadjuvant chemotherapy	29	NACT determined an increase in cfDI (*p* < 0.05) associated with tumor shrinkage and reduced Ki67 levels (*p* < 0.05). BC patients with pCR had a higher cfDI than patients with residual disease after surgery.
Lenher et al., 2013 [36]	Description of variation of cfDI during neoadjuvant chemotherapy	65	A downtrend of short and longer ALU amplicons from cycle one to six of neoadjuvant chemotherapy was observed in CR patients, whereas an increase was observed in non-responder (*p* = 0.033).
Deligezer et al., 2008 [35]	Description of variation of cfDI during adjuvant chemotherapy	41	CfDNA and cfDI varied simultaneously during adjuvant chemotherapy in BC patients.
Cheng et al., 2018 [38]	Prognostic value in metastatic BC	268	CfDI significantly increased in aBC patients after one cycle of chemotherapy (*p* = 0.00017 for ALU, *p* = 0.0016 for LINE-1). A higher cfDI (for both ALU and LINE-1) correlated with a higher PFS and OS.
Madhavan et al., 2014 [37]	Prognostic value in metastatic BC	283	cfDI (with ALU and LINE-1 amplicons) was lower in early BC patients (ALU: *p* = 0.046; LINE-1 *p* = 0.041). In aBC, lower values of cfDI were connected to a worse PFS (*p* = 0.0025 for ALU) and OS (*p* < 0.0001 for both ALU and LINE-1 fragments).

Another approach to determining the size of cfDNA fragments is based on electron microscopy [43]. However, neither electron microscopy nor qPCR are methods sufficient enough to elaborate a genome-wide study of cfDNA.

The advent of the next-generation sequencing (NGS) method has revealed the chance to use a single-base resolution assay to identify cfDNA. In some studies, cfDNA molecules were sequenced with a paired-end sequencing technique (specifically, bisulfite-based sequencing) [17]. The paired-end reads were lined up with a corresponding genome, to identify the peripheral ends determining the whole fragment size. The limit of this technique is that if the fragments are larger than the read sequence, the sequence located in the middle and not present in the sequencing outcomes might be lost [17]. Among the new techniques, shallow whole genome sequencing (sWGS) has proven to be a useful method for selecting shorter fragments of cfDNA from cancer patients [27]. Mouliere et al. developed a method to identify specific cfDNA fragment sizes and combine them with sWGS, to gain more information about ctDNA [27]. Among the shallow whole-genome techniques, nanopore sequencing may be useful in sequencing very short DNA fragments [44]. Some preclinical data showed that this method could extract multiple information, such as fragmentation, methylation and nucleosomal patterns of DNA from cancer cells from a single run, without the problems observed with commonly used bisulfite-based sequencing for long or very short cfDNA fragments [45,46]. The limitations of PCR and widely-used NGS approaches for the analysis of long fragments may also be overcome by other multiparametric methods such as microfluidic capillary electrophoresis [47]. However, studies on BC patients with these new techniques are awaited.

### 2.2. Nucleosomal Patterns

The size of cfDNA fragments is also determined by epigenetic processes. It is known that hypomethylated cfDNA fragments tend to be shorter than hypermethylated DNA [19,48,49]. The nucleosomal complex is influenced by methylation and, notably, in hypomethylated cfDNA the nucleosomes are fewer; consequently, the DNA is more susceptible to nucleases and the fragments may have a more variable size [50].

The majority of cfDNA fragments are constituted by mono-nucleosomal DNA, but longer fragments may present more nucleosomes (i.e., dinucleosomes, trinucleosomes) [51]. However longer fragments can be degraded to shorter cfDNA fragments, building up the mono-nucleosomal pool, and often tend be excluded by preanalytical procedures that work on the isolation of mono-nucleosomal cfDNA, with biased results [52,53]. Longer fragments often carry more mutational signatures that can get lost with only mono-nucleosomal analysis; instead, di/tri-nucleosomal analysis may increase the detection of ctDNA mutations [54,55].

In consideration of this, nucleosomal patterns may be the determinant for establishing the size and topology of circulating DNA fragments, and may also be informative about the cell of origin, and cell physiology or pathology [19]. The nucleosome occupancy in DNA is different among various tissues and particularly in cancer cells, compared with healthy cells [46]. In particular, the main protein units of the nucleosomal subunit are the histones, which undergo post-translational modifications (PTMs) [56]. These PTMs include acetylation, methylation, phosphorylation, ubiquitination, sumoylation, glycosylation, homocysteinylation and crotonylation [50]. These modifications do not alter the DNA sequence, but can influence the chromatin structure and consequently the production of different-sized fragments [57]. Moreover, the mechanisms of cfDNA release such as apoptosis or necrosis, influence the nucleosomal patterns [58,59]. The analysis of different PTMs proved to be a potential circulating biomarker for cancer. Their presence in the bloodstream was enhanced, due to the rapid cancer-cell turnover after chemotherapy treatments [60,61]. According to this hypothesis, the evaluation of the levels of circulating nucleosomes before and after treatment has been studied [50].

In BC patients undergoing neoadjuvant chemotherapy, it was observed that higher basal levels of nucleosomes corresponded to a worse response [62]. However, the majority of available data derive from preclinical studies, and we will hopefully experience the application of new techniques involving nucleosomes in the near future.

### 2.3. End-Fragments Signature

Recently, the presence of preferred ends in circulating fragments was demonstrated. The fragmentation of cfDNA molecules does not derive from a random process. Preferred ends of fragments refer to specific ending-sites in the genome, usually in regions with open chromatin (which are more accessible) [63]. Chromatin accessibility differs among various tissues, as do nucleosomal patterns [19]. Of note, it has been observed that there are specific ctDNA fragments end-coordinates among hepatocarcinoma patients, which could serve as biomarkers for cancer [64]. A study on multiple cancer types (including BC) showed that ends of ctDNA fragments tend to be located at different genomic loci, compared with ends of fragments produced by peripheral blood cells. Data from >2700 plasma-DNA samples from patients with 11 different tumors (including healthy controls) were analyzed with WGS, showing a 95% specificity and sensitivity of end signature-analysis for cancer detection, approximately 79% for all cancer types, and in particular 78% for BC; however, this study has not been peer reviewed [65]. If these data are reproduced, this method could become a possible cost-effective way to detect BC at an early stage, thanks to the wide presence of preferred end-coordinates across the genome.

In addition to preferred end size, the composition of end motifs has become increasingly interesting. The end motifs correspond to the number and type of nucleotides proximal to the 5’end of each fragment. Deoxyribonucleases (DNases) are the major actors in cutting fragments and determining the specific end-motifs. Each DNase cuts the fragment into different sequences, for example, DNase1 cleaves protein-free DNA, whereas DNase1L3 cuts chromatin into the nucleosomal linker sequences [66]. In several cancers, including BC, the level of DNase1L3 is lower than in normal cells, resulting in fewer DNase1L3-associated end motifs [67]. Studies have shown a possible clinical application for end motifs in cancer detection [68].

Finally, cDNA fragments were shown to present single-stranded 5′ DNA protrusions denominated “jagged ends”, traced thanks to the use of methylcytosine during DNA end-repair for WGS library development [69]. Approximately 88% of cfDNA fragments present jagged ends, and tumoral DNA fragments seem to have an increased “jaggedness” compared with normal cfDNA [17]. Their formation depends on different DNases activity and nucleosomal footprints [70]. Nowadays, there are few data about the use of jagged ends in cancer diagnosis. Plasma- and urinary-DNA jaggedness has been used for detecting patients with hepatocellular and bladder carcinoma, showing a potential application in oncology (i.e., a surrogate for the analysis of different DNase activities) [71,72]. Further evidence is awaited to investigate jaggedness in patients with BC and other different cancers.

### 2.4. Epigenomic Modifications of cfDNA Fragments: Methylation and Hydroxymethylation

Changes in epigenomic features are typically observed in cancer also in the early stages of the disease, motivating their analysis as a new tool for early diagnosis or prognostic biomarkers. The methylation profiling of cfDNA fragments is a new method for identifying patterns of cancer tissue [73,74]. Methylation patterns are unique to cancer cells, with hypermethylation for the silencing of tumor-suppressor genes and the hypomethylation of oncogene to enhance oncogenic activity. The realization of reference methylomes from normal cells and cancer cells allowed for the definition of new non-invasive methods to discriminate against cancer or other diseases [19].

Various techniques have been used to study methylation patterns. The “cell-free DNA methylation immunoprecipitation” (cfMeDIP-seq) method was used to find methylations on a genome-wide scale, using liquid biopsy to identify cancers that do not usually release useful amounts of ctDNA (such as renal cancer of intracranial tumors) [75,76].

Regarding BC, “The Cancer Genome Atlas” (TCGA) network identified five subgroups of BC which were different from each other, based on DNA-methylation patterns, with one of them presenting a CpG island methylator phenotype (CIMP) [77]. Some clinical trials have started, to determine a potential use for methylation in BC (Table 2).

In a study conducted by Liu et al. involving 80 patients with malignant BC and 80 patients with a benign breast-tumor, the analysis of differentially methylated regions of cfDNA was combined with other fragment information (such as size) and radiologic information, to anticipate the detection of malignant BC [78]. The cfDNA of each patient was studied with whole-genome bisulfite sequencing (WGBS), highlighting the fact that the quantity of cfDNA in different genomic regions is negatively related to their CpG density (genomic sites which are usually methylated) in cancer patients [78]. Of note, in another study it was observed that the methylation profile of cfDNA can change years before BC is clinically identified [79].

“CMethDNA assay” is a further method that potentially retains more CpG-rich informative fragments [74]. This method has been used to apply methylation information to predict survival outcomes in advanced BC patients [80]. It is a 10-gene panel of cfDNA methylation markers using a quantitative PCR assay, derived from known BC hypermethylated genes identified from DNA-methylation patterns in breast tissue (such as methylated RASSF1A, APC and BRCA1) [80,81]. Using this technique, Visvanathan et al. tested methylation patterns in 141 metastatic BC patients at the baseline, 4 weeks after treatment initiation and at the first radiological disease-assessment. There was a significant association between an elevated cumulative-methylation-index (CMI), and both a shorter PFS and OS after 4 weeks of treatment [82]. Panagopoulou et al. used a similar method consisting of capillary electrophoresis (a multiparametric method). [83]. They considered five other frequently methylated genes (KLK10, SOX17, WNT5A, MSH2, GATA3) in the cfDNA of BC patients, which were found to be more methylated than those in the healthy controls [83]. In the same study, the authors observed simultaneously many features of cfDNA fragments, from methylation patterns to size (with a SYBR Green-based/Qubit assay), enhancing a more fragmented pattern in larger tumors. They also found a correlation between the quantification of cfDNA with a worse PFS, higher death risk and non-response to pharmacotherapy, in particular in the metastatic setting [83].

**Table 2 ijms-23-14197-t002:** Registered clinical trials assessing methylation patterns in cfDNA in breast cancer.

Trial	Objective	Patients Enrolled (n)	Type	Status or Available Results
NCT03480659	Early detection	400	Observational	Terminated (technical problem with plasma blood samples obtained from the patients).
NCT03863522 Visvanathan et al., 2017 [82]	Early detection	447	Interventional	Completed: elevated cumulative methylation index is significantly associated with both shorter PFS and OS after 4 weeks of treatment in advanced BC.
Agostini et al., 2012 [29]	Diagnosis and early detection	39	Observational	Completed: cfDNA methylatation (RASSF1A, MAL and SFRP1) is a phenotypic feature of BC, and can be used for cancer diagnosis.
NCT03184090	Prediction of treatment benefit	33	Interventional	Completed, no result posted.
NCT04996836	Prediction of treatment benefit	200	Observational	Active, not recruiting yet.
NCT00698477	Prediction of treatment benefit and mechanism of resistance	30	Observational	Unknown.
NCT03205761	Prediction of treatment benefit	34	Interventional	Active, not recruiting.

Methylation patterns have also been evaluated as biomarkers for the prediction of treatment response [84]. In a study conducted by Zurita et al., the serum levels of methylated gene-promoter 14-3-3-σ were observed in 34 metastatic BC patients during chemotherapy, and 77 patients previously treated for BC [84]. In the study, the variation of the methylation was considered the “biomarker response ratio”, and connected to response and prognosis. The power of methylation analysis in predicting treatment response has also been studied by Fackler et al. and Legendre et al. who, with different techniques (such as cMethDNA) hypothesized an application in therapy stratification or the identification of recurrent metastatic BC [80,85].

Furthermore the oxidation of 5mC, catalyzed by the ten-eleven translocation enzymes (TETs), is considered one of the active mechanisms for DNA demethylation, obtaining 5-hydroxymethylcytosine (5hmC) [86]. The presence of low quantities of 5hmC has been observed in different cancers as a consequence of altered TETs activity [87]. The analysis of this epigenetic alteration is under study for cancer diagnosis and stage classification, but there are still not many data available on BC [88]. Recently, it has also been demonstrated that 5hmC variations occur first in tumorigenesis, and tend not to change during cancer progression at the various stages [89].

Despite the intriguing results, these findings need further validation to determine the clinical utility of methylation analysis in BC.

## 3. Limitations and Critical Issues

First of all, one of the major critical issues for the use of fragment analysis in BC is the wide discrepancy among authors concerning ctDNA-fragment length. Some studies assume that ctDNA is made of longer amplicons, compared with healthy cfDNA, while others suggest the complete opposite [23]. This debate also reflects on the value of cfDI. Most of the studies presented assume that a higher cfDI is related to the presence of BC, higher stages, and worse outcome after treatments [26,33,34,35]. However, this is not verified in the advanced setting, where the opposite was demonstrated [37,38]. As a clear example, Umetani et al. proposed that an increased cfDI could be a prediction of metastasization, but afterwards Cheng et al. demonstrated that cfDI was lower in BC patients developing a recurrence [26,38]. The reason for these controversial results may derive from the different methods used for evaluating cfDI considering different fragments (e.g., ALU sequences and others), and the different population of patients taken into account.

It is well known that the pre-analytical collection, processing of blood samples, centrifugation protocols and long-term storage can have significant quantitative and qualitative effects on cfDNA (such as hematopoietic lysis, cfDNA damage and contamination) [31,90]. Methods of DNA extraction, different kits and the use of plasma versus serum in the different studies may influence the results, with different amounts of cfDNA analyzed. Moreover the intrinsic limits of qPCR (the most used technique in studies on BC and cfDI) are probably also responsible.

Not less important are the small sample sizes of the studies available and the necessity for wide prospective-validation cohorts. Another consideration concerns the necessity for standard reference-markers to adequately compare studies [91]. Actually, there is no agreement on standardized methods for pre-analytical studies, and this reduces the strength of the body of evidence.

To date, there is still a scarcity of application of some new techniques derived from genome sequencing, and research in oncology into some aspects of fragmentomics (i.e., jagged end, end motif) is only at the beginning (compared with other application fields such as transplantology and noninvasive prenatal diagnostics) [17].

Finally, the most important aspect in transferring all these findings to BC clinical practice will be the acquisition of analytical validity (from an evaluation of accuracy, reliability and reproducibility), and clinical validity (the power to identify populations with significantly specific clinical-outcomes) from studies that are still overdue in this field [51].

## 4. Conclusions

The advent of overwhelming new data from “omics” in cancer diagnosis will progressively change the therapeutic and diagnostic approach in cancer. Fragmentomics analysis, in particular fragment size and cfDI, is establishing itself as a non-invasive, cost-effective new informative method in BC. In particular, it will change the approach to liquid biopsy, gaining helpful information independently from the mutational signature. Epigenetic features are more dynamic, and global fragmentation-patterns could be combined as different biomarkers, thanks to new bioinformatic techniques (i.e., DNA evaluation of fragments for early interception, DELFI) to reflect BC disease [22].

The potential applications seem to be numerous, from cancer detection and the anticipation of recurrence and the evaluation of minimal residual disease, to responsiveness or resistance to treatments. However, the gap between research and clinical practice is still deep, and larger studies are needed to fill in the missing points. Further results from fragmentomics in BC are awaited, and we look forward to new opportunities for personalized cancer treatment.

## Figures and Tables

**Figure 1 ijms-23-14197-f001:**
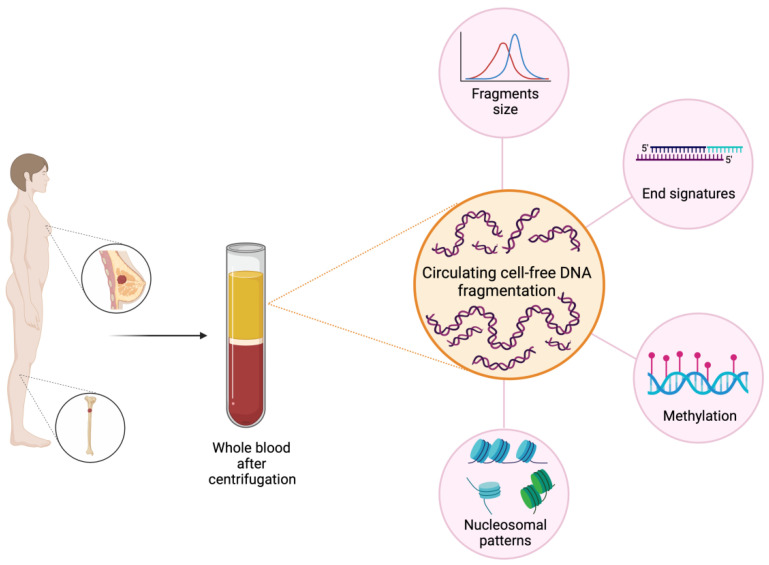
Fragmentomics features in BC. The study of fragmentomics in BC includes various aspects such as the evaluation of fragment size and cfDI, the study of fragment ends with specific motifs included in the sequence of the fragment and the nucleosome footprints or position. The potentiality of fragmentomics covers a wide range of applications in BC, from the early stage with a role in cancer detection and evaluation of the minimal residual disease, to advanced stages, where it may be useful in the prediction of treatment efficacy or resistance and the evaluation of response to therapies. Created with BioRender.com.

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
