# Peer review of "Cell-Free DNA Fragmentomics: A Promising Biomarker for Diagnosis, Prognosis and Prediction of Response in Breast Cancer"

_ijms, 2022, doi:10.3390/ijms232214197_

Round 1

Reviewer 1 Report

Please see the attached document

Author Response

Dear reviewer thanks for your precious suggestions.

  1. all the spelling and grammar errors have been corrected.
  2. As referred to section 2.1: limits about pre-analytical variables are mentioned in the section "limits and critical issues",  the variability of length of tumoral DNA fragments has been highlighted in the paragraph " CfDNA fragmentomics in BC: different features and applications",  the references suggested have been incorporated in the text mentioning the new analysis approaches proposed.
  3. Section 2.2: lines 170-171, a mention to original studies has been added
  4. Section 2.3: clarifications about nucleosomal analysis have been added, including the suggested references

Reviewer 2 Report

Gianni et al

Cell-free DNA fragmentomics: a promising biomarker of response in breast cancer

General remarks

a fairly good overview on the topic with some points missing (see below)

specific comments

since chapter 2.4 makes a significant part of the paper the title of the MS should be changed respectively and epigenomic changes included

lines 94-95

The ratio between shorter and longer fragments is defined as the cfDNA integrity index (cfDI) and gives information about the total amount of tumoral DNA fragments.

This might be true when the majority of longer or shorter fragments are tumor derived but this is not really clear so far. There is still a debate on the question whether longer or shorter fragments originate from tumor cells and therefore I would use a different wording. The authors confirm the need to answer these questions by referring to several papers which describe opposite results (line 114ff).

Line 240ff

this sentence is incomplete and needs to be reworded

line 302ff

the same is true for the influence of different isolation procedures (kits) on the amount/composition of cfDNA applied in different studies, in addition the use of plasma vs. serum in different papers needs to be considered, as well as the fact that there is no agreement on a standardized method for the pre-analytical handling of blood samples, these points should be mentioned in a more detailed way,

Author Response

Dear reviewer, thank for your suggestions.

  1. We considered the epigenetic modification of DNA as part of the multiple fragment features, we displayed that in figure 1, so we do not consider to change the title or the paper or the title of the macro section;
  2. lines 94-95. Clarifications about the ongoing debate have been modified in the text;
  3. line 240 has been removed;
  4. line 302. We provide insights about the limitations in the new version of the text as suggested.
  5. English review has been provided by a professional translator

Reviewer 3 Report

This is an excellent review and most relevant to the field, which I read with great interest. The only significant suggestion I would have is to put some more emphasis in describing the multiple methodologies used by different groups producing these results, highlighting the great significance of standardized pre-analytical processes and SOPs in order to reach comprehensive conclusions. I would also suggest a change on the title to a broader one, as the review does not focus only in the value for prediction of response but rather diagnosis, prognosis and prediction of BC. The information included in the already referenced article of Panagopoulou et al., in Oncogene, regarding results of cfDNA capillary electrophoresis from BC patients and controls, which show some statistical differences confirming a more fragmented pattern in larger tumors could be added in the relevant section. In addition, the article of Panagopoulou et al., J Cell Physiol presents the fragmentation cfDNA pattern of human breast cancer MCF-7 cells under treatment.

Minor

Line 93 Ref 23 rather than 123.

Line 57 delete dot.

127 ‘distant metastasis comparison after surgery’ Rephrase

172 join paragraph.

202 ‘fragment end’

207  ‘…showing a 95% specificity and sensitivity for cancer detection’ of what?

209 join paragraph and add in the beginning of the sentence: ‘ If these data are reproduced…”

240-243 This sentence should be incorporated in the paragraph above and I believe this is not the correct reference to it.

250 Replace ‘studies’ with ‘clinical trials’.

295 add dot

Author Response

Dear reviewer, thank for your suggestions.

We propose a new title as suggested and we have completed the text as recommended.

English has been revised and all the minor revisions have been done.
